# Empirical Study on Fluctuation Theorem for Volatility Cascade Processes in Stock Markets

**DOI:** 10.3390/e27040435

**Published:** 2025-04-17

**Authors:** Jun-ichi Maskawa

**Affiliations:** Department of Economics, Seijo University, 6-1-20, Seijo, Setagaya-ku, Tokyo 157-8511, Japan; maskawa@seijo.ac.jp

**Keywords:** econophysics, intermittency, multifractality, financial time series, Langevin equation, stochastic thermodynamics, integral fluctuation theorem

## Abstract

This study investigates the properties of financial markets that arise from the multi-scale structure of volatility, particularly intermittency, by employing robust theoretical tools from nonequilibrium thermodynamics. Intermittency in velocity fields along spatial and temporal axes is a well-known phenomenon in developed turbulence, with extensive research dedicated to its structures and underlying mechanisms. In turbulence, such intermittency is explained through energy cascades, where energy injected at macroscopic scales is transferred to microscopic scales. Similarly, analogous cascade processes have been proposed to explain the intermittency observed in financial time series. In this work, we model volatility cascade processes in the stock market by applying the framework of stochastic thermodynamics to a Langevin system that describes the dynamics. We introduce thermodynamic concepts such as temperature, heat, work, and entropy into the analysis of financial markets. This framework allows for a detailed investigation of individual trajectories of volatility cascades across longer to shorter time scales. Further, we conduct an empirical study primarily using the normalized average of intraday logarithmic stock prices of the constituent stocks in the FTSE 100 Index listed on the London Stock Exchange (LSE), along with two additional data sets from the Tokyo Stock Exchange (TSE). Our Langevin-based model successfully reproduces the empirical distribution of volatility—defined as the absolute value of the wavelet coefficients across time scales—and the cascade trajectories satisfy the Integral Fluctuation Theorem associated with entropy production. A detailed analysis of the cascade trajectories reveals that, for the LSE data set, volatility cascades from larger to smaller time scales occur in a causal manner along the temporal axis, consistent with known stylized facts of financial time series. In contrast, for the two data sets from the TSE, while similar behavior is observed at smaller time scales, anti-causal behavior emerges at longer time scales.

## 1. Introduction

We often encounter cases where seemingly different complex systems, composed of different elements and governed by different dynamics, exhibit common prominent features. The intermittency observed in developed turbulence and volatility clustering seen in financial time series of the stock market are examples of such phenomena.

Intermittency in turbulence is characterized by sudden temporal changes in the statistical characteristics of fluctuations in the velocity difference between two spatial points and the spatial coexistence of large and small fluctuations [1]. On the other hand, volatility clustering is considered a stylized fact universally observed in financial market price time series, where bursts of volatility occur irregularly and form clusters within otherwise calm price fluctuations [2,3]. The GARCH class of models have been proposed as a time series model capable of reproducing volatility clustering [4,5]. The phenomenon also manifests as long-range dependence in the autocorrelation of volatility. To explain this phenomenon, models based on the FIGARCH class in discrete time [6], as well as continuous-time models employing fractional Brownian motion [7], have been proposed. Ongoing research continues to elaborate on these models and discuss their characterization through the Hurst exponent [8,9,10,11,12]. At each time scale, intermittency and volatility clustering produce a characteristic hierarchical structure known as multifractality, which is characterized by the self-similar transformation rule of the probability density function of price fluctuations and the nonlinear scaling law of the structure function (the nth moment of fluctuations) [13,14,15].

The key idea in reproducing multifractal time series is the construction of a multifractal measure along the time axis [16,17,18,19]. A leading model that exhibits multifractality is the cascade model. In developed turbulence, the process by which mechanically generated vortices on a macroscopic scale deform and destabilize according to the Navier–Stokes equation, eventually splitting into smaller vortices, is considered an energy cascade [13,20,21,22]. Similarly, modeling multifractal time series using a recursively multiplicative random cascade process from a coarse-grained scale to a microscopic scale has provided an effective means to describe financial time series [23,24,25,26,27,28,29].

On the other hand, there have also been studies that directly observe cascade processes and aim to model them. It has been proposed that the multiplicative random cascade process observed in fluid dynamics and financial markets can be regarded as a Markov process and described by the Fokker–Planck equation or the Langevin equation [30,31,32,33,34,35].

The Langevin system has been extensively studied within the framework of stochastic thermodynamics, a field that has seen remarkable advancement, leading to a deeper thermodynamic understanding of such systems [36,37,38,39,40,41,42]. An important advancement that uses stochastic thermodynamics effectively has been achieved in the field of developed turbulence. Nickelson and Engel demonstrated the Integral Fluctuation Theorem, a fundamental theorem in the field of stochastic thermodynamics, in developed turbulent systems [43,44]. This provides a justification for modeling cascade processes using Markovian approaches, including the Langevin and Fokker–Planck equations.

In this study, by applying the stochastic thermodynamic framework to the Langevin system proposed in [35], we incorporate thermodynamic concepts such as temperature, heat, work, and entropy into financial market analysis. The approach adopted in [35], which analyzes a Langevin system through the corresponding Fokker–Planck equation, is well suited to capture the statistical properties of volatility at each time scale. In contrast, our description based on the Langevin equation enables a detailed analysis of the individual trajectories of volatility cascade processes from longer to shorter time scales. We also conduct an empirical study primarily using the normalized average of intraday logarithmic stock prices of the constituent stocks in the FTSE 100 Index listed on the London Stock Exchange (LSE), along with two additional data sets from the Tokyo Stock Exchange (TSE). Our Langevin-based model successfully reproduces the empirical distribution of volatility—defined as the absolute value of the wavelet coefficients across time scales—and the cascade trajectories satisfy an Integral Fluctuation Theorem associated with entropy production.

In financial time series, past coarse-grained volatility measures have been found to correlate more strongly with future fine-scale volatility than the reverse process. This causal structure of financial time series was first reported by Müller et al. [45]. Since then, the causal structure between time scales, i.e., the flow of information from long- to short-term scales, has been empirically investigated in financial markets and supported by multiple studies as a stylized fact of financial time series [46,47]. We investigated the causal structure of the trajectories employing the method developed herein to track individual volatility cascade processes. A detailed analysis of the cascade trajectories reveals that, for the LSE data set, volatility cascades from larger to smaller time scales occur in a causal manner along the temporal axis, consistent with known stylized facts of financial time series. In contrast, for the two data sets from the TSE, while similar behavior is observed at smaller time scales, anti-causal behavior emerges at longer time scales.

## 2. Materials and Methods

### 2.1. Multiplicative Random Cascade Model of Volatility

Let us define the volatility field v(t,s) as a function in the upper half-plane, with time *t* and time scale *s* along the axes:(1)$$S^{+}=\left\{\left(t,s\right)\in R^{2}|s>0\right\}$$
An example of such volatility is the empirical volatility of scale *s*, defined for a price time series P(t) as(2)v(t,s)=|logP(t+s)−logP(t)|Next, consider a trajectory t(s) within $S^{+}$ and define the volatility along the trajectory as x(s)=v(t(s),s). Following [35], we consider the following stochastic differential equation (SDE) for the volatility x(s)=v(t(s),s):(3)$$dx\left(\lambda \right)=x\left(\lambda \right)\circ \left(-\gamma_{M}d\lambda +\sigma_{M}dB_{M}\left(\lambda \right)\right)+a_{A}\left(\lambda \right)d\lambda +b_{A}\left(\lambda \right)dB_{A}\left(\lambda \right).$$Here, λ=logL/s, where *L* is an appropriately chosen macroscopic scale parameter, taken as the length of the time series used in the empirical study of this article. Furthermore, $dB_{M}\left(\lambda \right)$ and $dB_{A}\left(\lambda \right)$ are independent Brownian motions. The parameters $\gamma_{M}$ and $\sigma_{M}$, as well as the functions $a_{A}\left(\lambda \right)$ and $b_{A}\left(\lambda \right)$, are estimated from the time series data, as described later. Note that, in this study, the product between the function x(λ) and the Brownian motion dB(λ) follows the Stratonovich product (denoted by ∘). In [35], the Itô product was used. The Stratonovich product between the system variable *x* and the Brownian motion $dB_{M}$ can be expressed using the Itô product (denoted by ·) as follows [48]:(4)$$x\left(\lambda \right)\circ dB_{M}\left(\lambda \right)=x\left(\lambda \right)\cdot dB_{M}\left(\lambda \right)+\frac{\sigma_{M}}{2}d\lambda .$$By replacing $\gamma_{M}$ with $\gamma_{M}-\sigma_{M}^{2}/2$ in the stochastic differential equation using the Itô product as Equation (Equation 7) in [35], the SDE (Equation 3) can be obtained. The right-hand side of the SDE (Equation 3) consists of the sum of two types of terms: drift and diffusion terms. The former expressed as a multiplication with the stochastic variable *x*, and the latter is expressed as an addition.

The dominant multiplicative terms represent the cascade of volatility *x* along the time scale axis, and the solution to the SDE,(5)$$dw\left(\lambda \right)=w\left(\lambda \right)\circ (-\gamma_{M}d\lambda +\sigma_{M}dB_{M}\left(\lambda \right))$$
is expressed as(6)$$w\left(\lambda \right)=w\left(0\right)\cdot \exp\left\{-\gamma_{M}\lambda +\sigma_{M}B_{M}\right\}.$$When the functions $a_{A}\left(\lambda \right)$ and $b_{A}\left(\lambda \right)$ are considered as infinitesimal quantities compared with the dominant terms, the solution w(λ) can be regarded as the zeroth-order approximation of x(λ).

The power law behavior of the *q*-th moment $E[w{\left(\lambda \right)}^{q}]$ (*q*-th structure function) as a function of scale *s* is proven by solution (Equation 6) as follows:(7)$$\begin{array}{cc} E[w{\left(\lambda \right)}^{q}] & \;=E\left[w{\left(0\right)}^{q}\right]\exp\left\{-\gamma_{M}\lambda q+\frac{1}{2}\sigma^{2}\lambda q^{2}\right\}\\ \; & \;=E\left[w{\left(0\right)}^{q}\right]{\left(\frac{s}{L}\right)}^{\gamma_{M}q-\frac{1}{2}\sigma^{2}q^{2}} \end{array}$$
showing the multifractality of the signal v(s,t) because the scaling exponent $\zeta \left(q\right)=\gamma_{M}q-\frac{1}{2}\sigma^{2}q^{2}$ is a convex upward nonlinear function.

### 2.2. Parameter Estimation

From Equation (Equation 3), the time evolution equations for the mean E[x] and variance $V\left[x\right]=E\left[x^{2}\right]-E{\left[x\right]}^{2}$ of the stochastic variable *x* can be derived as follows:(8)$$\begin{array}{c} \;\;\;\;\;\;\;\;\;\;\;\;\;\;\;\;\;\;\;\;\;\;\;\;\;\;\;\;\;\;\;\;\;\;\;\;\;\;\;\;\;\;\;\;\;\;\;\;\;\;\;\;\;\;\;\;\;\;\;\;\frac{dE[x]}{d\lambda }=-\left(\gamma_{M}-\sigma_{M}^{2}/2\right)E\left[x\right]+a_{A}\left(\lambda \right) \end{array}$$(9)$$\begin{array}{c} \frac{dV[x]}{d\lambda }=-2\left(\gamma_{M}-\sigma_{M}^{2}/2\right)\left(V\left[x\right]+E{\left[x\right]}^{2}\right)+2a_{A}E\left[x\right]+b_{A}^{2}-2E\left[x\right]\frac{dE[x]}{d\lambda } \end{array}$$

In an empirical study of the LSE data set conducted in [35], it was shown that the mean E[x] and the variance V[x] evolve over time following the same power law (Figure 1 in [35]). Here, assuming $E\left[x\right]=\sqrt{V[x]}=As^{0.5}$, it follows from the above equation that the functions $a_{A}\left(\lambda \right)$ and $b_{A}\left(\lambda \right)$ are power functions with the same exponent. Furthermore, two relationships can be derived as follows:(10)$$\begin{array}{c} a_{A}\left(\lambda \right)=as{\left(\lambda \right)}^{0.5} \end{array}$$(11)$$\begin{array}{c} b_{A}\left(\lambda \right)=bs{\left(\lambda \right)}^{0.5} \end{array}$$
and, therefore,(12)$$\begin{array}{c} a=(\gamma_{M}-\sigma_{M}^{2}/2-0.5)A \end{array}$$(13)$$\begin{array}{c} \;\;\;\;\;\;\;\;\;\;\;\;\;\;\;\;\;\;\;\;\;\;\;\;b=\sqrt{2Aa}. \end{array}$$In this study, for the LSE data set, the parameters $\gamma_{M}$, $\sigma_{M}$, *a*, and *b* are estimated using the approximate expression (Equation 7) of the *q*-th structure function for x(λ); the relations (Equation 12) and (Equation 13); and Equation (Equation 33), which is mentioned later. In the analysis of the TSE data set, we use the general relations (A3)∼(A6) (see Appendix A).

### 2.3. Integral Fluctuation Theorem

In this study, the volatility cascade processes in the stock market are studied within the framework of stochastic thermodynamics by considering them to be a Langevin system. Specifically, it is shown through an empirical study that it is possible to fix the parameters so that entropy production along appropriately defined trajectories within the upper half-plane $S^{+}$ satisfies an Integral Fluctuation Theorem [39,42].

The SDE (Equation 3) is defined as the time scale evolution equation satisfied by the stochastic variable x(s)=v(t(s),s) along a trajectory t(s) within $S^{+}$. Here, the SDE (Equation 3) is interpreted by analogy with the following overdamped Langevin equation, which describes the motion of colloidal particles in water. A typical overdamped Langevin equation is given by(14)$$\gamma \frac{dx(t)}{dt}=F\left(x\left(t\right),t\right))+\sqrt{2\gamma k_{B}T}\xi \left(t\right)$$
where γ is the viscous friction coefficient between the particle and water, *F* is the sum of the conservative force represented as the derivative of the potential *V* and the external force *f* applied directly, *T* is the temperature of the water (thermal bath), and ξ(t) represents the fluctuating force.

Using the relationship between the fluctuating force ξ(t) and the Brownian motion B(t),(15)$$\int_{0}^{t}\xi \left(t\right)dt=B\left(t\right)$$
we obtain the SDE(16)$$\gamma dx\left(t\right)=F\left(x\left(t\right),t\right))dt+\sqrt{2\gamma k_{B}T}dB\left(t\right)$$By comparing this equation with our model (Equation 3), the force F(x) acting on the system can be expressed as(17)$$\frac{F(x)}{\gamma }=-\gamma_{M}x+a_{A}$$Furthermore, it can be interpreted that the system receives fluctuating forces, $dB_{M}$ and $dB_{A}$, from heat baths with the following respective temperatures:(18)$$\begin{array}{c} \sqrt{\frac{2k_{B}T_{M}}{\gamma }}=\sigma_{M}x, \end{array}$$(19)$$\begin{array}{c} \sqrt{\frac{2k_{B}T_{A}}{\gamma }}=b_{A}\left(\lambda \right). \end{array}$$

Letting the variable $f(\lambda_{n})$ at the time scale $s_{n}=L\exp\left(-\lambda_{n}\right)$(n=0,⋯,N) be written as $f_{n}$, if we choose $\Delta \lambda_{n}=\Delta \lambda =const.$, the discretized SDE can be written as(20)$$\Delta x_{n}=x_{n+1}-x_{n}=\frac{F_{n}}{\gamma }\Delta \lambda +\sigma_{M}\frac{x_{n+1}+x_{n}}{2}\Delta B_{Mn}+b_{An}\Delta B_{An},$$
where(21)$$\frac{F_{n}}{\gamma }=-\gamma_{M}x_{n}+a_{An}.$$

The fluctuating force $\Delta B_{Xn}\left(X=M,A\right)$ follows a Gaussian distribution with a mean of 0 and variance of Δλ. Therefore, using the composition rule of Gaussian variables, the transition probability from state $x_{n}$ to $x_{n+1}$ is given by(22)$$P\left(x_{n+1}|x_{n};\lambda_{n}\right)=\frac{1}{\sqrt{2\pi \Delta \lambda \Sigma_{Mn}^{2}}}\exp\left(-\frac{1}{2\Delta \lambda \Sigma_{Mn}^{2}}{\left(x_{n+1}-x_{n}-\frac{F_{n}}{\gamma }\Delta \lambda \right)}^{2}\right),$$
where the quantity(23)$$\Sigma_{Mn}^{2}=\sigma_{M}^{2}{\left(\frac{x_{n+1}+x_{n}}{2}\right)}^{2}+{\left(\frac{b_{A(n+1)}+b_{An}}{2}\right)}^{2}$$
acts as the effective common temperature $2k_{B}T/\gamma$ of the two heat baths. Similarly, the transition probability for the reverse process from $x_{n+1}$ to $x_{n}$ is given by(24)$$P\left(x_{n}|x_{n+1};\lambda_{n}\right)=\frac{1}{\sqrt{2\pi \Delta \lambda \Sigma_{Mn}^{2}}}\exp\left(-\frac{1}{2\Delta \lambda \Sigma_{Mn}^{2}}{\left(x_{n}-x_{n+1}-\frac{F_{n+1}}{\gamma }\Delta \lambda \right)}^{2}\right)$$

The ratio of the transition probabilities in the reverse and forward directions can be written as follows:(25)$$\frac{P(x_{n}|x_{n+1};\lambda_{n})}{P(x_{n+1}|x_{n};\lambda_{n})}=e^{\beta_{n}\hat{Q}\left(x_{n+1},x_{n}\right)}$$
where(26)$$\gamma \beta_{n}=\frac{\gamma }{k_{B}T}=\frac{2}{\Sigma_{Mn}^{2}}$$
and $\hat{Q}\left(x_{n+1},x_{n}\right)$ represents the amount of heat transferred from the heat bath to the system during the transition from state $x_{n}$ to $x_{n+1}$. It is given by(27)$$\hat{Q}\left(x_{n+1},x_{n}\right)=-\left(F_{n+1}+F_{n}\right)\frac{x_{n+1}-x_{n}}{2}.$$

When the initial state of the system in the forward process is $P(x,\lambda_{i})$ and the final state is $P(x,\lambda_{f})$, the probability of the forward process $x_{\lambda_{f}}={\left\{x\left(\lambda \right)\right\}}_{\lambda =\lambda_{i}}^{\lambda_{f}}$ occurring is given by(28)$$P\left(x_{\lambda_{f}}\right)=\left(\prod_{n=0}^{N-1}P\left(x_{n+1}|x_{n};\lambda_{n}\right)\right)P\left(x,\lambda_{i}\right)$$On the other hand, the probability of the reverse process $x_{\lambda_{f}}^{†}={\left\{x\left(\lambda \right)\right\}}_{\lambda =\lambda_{f}}^{\lambda_{i}}$ occurring is given by(29)$$P\left(x_{\lambda_{f}}^{†}\right)=\left(\prod_{n=0}^{N-1}P\left(x_{n}|x_{n+1};\lambda_{n}\right)\right)P\left(x,\lambda_{f}\right)$$
where $x_{0}=x\left(\lambda_{0}=\lambda_{i}\right)$ and $x_{N}=x\left(\lambda_{N}=\lambda_{f}\right)$. Using Equation (Equation 25), the ratio of the path probabilities of the reverse and forward processes is given by(30)$$\frac{P(x_{\lambda_{f}}^{†})}{P(x_{\lambda_{f}})}=\frac{P(x,\lambda_{f})}{P(x,\lambda_{i})}e^{\sum_{n=0}^{N-1}\beta_{n}\hat{Q}\left(x_{n+1},x_{n}\right)}=e^{-\hat{\sigma }\left[x_{\lambda_{f}}\right]}$$
where $\hat{\sigma [x_{\lambda_{f}}]}$ represents the total entropy production of the entire system, including the heat bath, during the forward process $x_{\lambda_{f}}$, and is given by(31)$$\hat{\sigma }\left[x_{\lambda_{f}}\right]=-\log\left(\frac{P(x,\lambda_{f})}{P(x,\lambda_{i})}\right)-\sum_{n=0}^{N-1}\beta_{n}\hat{Q}\left(x_{n+1},x_{n}\right)$$Equation (Equation 30) is called the Detailed Fluctuation Theorem. In Equation (Equation 31), the first term represents the change in the Shannon entropy of the system described by the state *x*, which is given by(32)$$\hat{\sigma_{x}}\left[x_{\lambda_{f}}\right]=-\left(\log P\left(x,\lambda_{f}\right)-\log P\left(x,\lambda_{i}\right)\right).$$The second term represents the entropy change in the heat bath as a result of heat influx. By using Equation (Equation 30) and taking the expectation value of $\exp(-\hat{\sigma }\left[x_{\lambda_{f}}\right])$ over all possible forward trajectories, the Integral Fluctuation Theorem can be derived as follows:(33)$$E[e^{-\hat{\sigma }\left[x_{\lambda_{f}}\right]}]=1.$$

### 2.4. Empirical Study

#### 2.4.1. Data

In this study, we analyzed three data sets (1, 2, and 3).

#### 2.4.2. Data Processing

##### Data Set 1 (LSE Data Set: Normalized Average)

We primarily analyzed the normalized average of the logarithmic stock prices of the constituent stocks of the FTSE 100 Index, listed on the LSE, from November 2007 to January 2009. This period includes the Lehman shock on 15 September 2008, and the market crash on 8 October 2008. The constituents of the FTSE 100 Index are regularly updated. We selected $N_{F}=111$ stocks that remained listed on the LSE throughout this period. We excluded the overnight price change and specifically examined the intraday evolutions of returns. We set δt=1 (min) and examined the log-return with a time interval of 10 min or more to avoid microstructure noise. The following steps were taken to prepare the data required for this empirical study:From the one-minute price data $S_{i}\left(t\right)\left(i=1,\cdots ,N_{F}\right)$ of each stock’s intraday price fluctuations, the logarithmic return $\delta R_{i}\left(t\right)=\log\left(S_{i}\left(t\right)\right)-\log\left(S_{i}\left(t-\delta t\right)\right)$ was calculated.The normalized average of the logarithmic return was computed as(34)$$\delta R\left(k\delta t\right)=\frac{1}{N_{F}}\sum_{i=1}^{N_{F}}\frac{\delta R_{i}\left(k\delta t\right)-\mu_{i}}{\sigma_{i}}.$$
To remove the effect of intraday U-shaped patterns of market activity from the time series, the return was divided by the standard deviation of the corresponding time of day for each issue *i*.We cumulated δZ(t) to obtain the path of the process Z(kδt)(k=1,⋯,L) as follows:(35)$$Z\left(k\delta t\right)=\sum_{k^{\prime }=1}^{k}\delta Z\left(k^{\prime }\delta t\right).$$

##### Data Set 2 (TSE Data Set: Normalized Average)

We analyzed two additional data sets from the TSE. To investigate differences between stock markets, we analyzed the normalized average of logarithmic stock prices for the constituent stocks of the NIKKEI 225 Index listed on the TSE, covering the period from October 2007 to December 2009. Price changes during the overnight period and the lunchtime intermission were excluded, allowing us to focus specifically on the intraday evolution of returns. The data processing steps were the same as for the LSE data set.

##### Data Set 3 (TSE Data Set: Individual Stocks)

To investigate how market conditions affect results, we further analyzed the price of individual TSE-listed stocks during the post-crisis period from January 2014 to December 2015, following the financial turmoil of 2007–2010. We chose the seven highly traded stocks that represent the industry among the constituent stocks of the NIKKEI 225 Index, that is, Mitsubishi UFJ Financial Group, Inc.; SoftBank Group Corp.; Mitsubishi Corporation; Komatsu Ltd.; Toyota Motor Corporation; Mizuho Financial Group, Inc.; and Nomura Holdings, Inc. The headquarters of these companies are located in Tokyo, Japan. For these data sets, price changes during the overnight period and lunchtime intermission were also excluded.

#### 2.4.3. Wavelet Transform

These analyses examined the following wavelet transform of the normalized average of the logarithmic stock prices denoted by Z(t),t∈[0,L]:(36)$$W_{\psi }Z\left[u,s\right]=\int_{-\infty }^{+\infty }Z\left(t\right)\frac{1}{s}\psi^{*}\left(\frac{t-u}{s}\right)dt,\;u\in \left[0,L\right],$$
where the function ψ is designated as the analyzing wavelet [49,50]. When using the delta function ψ(t)=δ(t−1)−δ(t) as the analyzing wavelet, the wavelet transform $W_{\psi }Z\left[u,s\right]=Z\left(u+s\right)-Z\left(u\right)$ is exactly the logarithmic return of the period *s*. Here, we use the second derivative of the Gaussian functions as follows:(37)$$\psi \left(t\right)=\frac{d^{2}}{dt^{2}}\left(e^{-\frac{t^{2}}{2}}\right)=\left(t^{2}-1\right)e^{-\frac{t^{2}}{2}}.$$
In general, by using the *n*-th derivative of the function with asymptotic fast decay as the analyzing wavelet, one can remove the local trend of the *m*-th order (m≤n−1) because the function is orthogonal to *m*-th order polynomials. For the second derivative of the Gaussian functions, the linear trends of Z(t) with scale *s* are eliminated in the wavelet transform $W_{\psi }Z\left[u,s\right]$.

In the actual financial market, the price fluctuation is non-stationary and the volatility is not observable. The quantity used herein is the absolute value of the wavelet transform $x\left(\lambda \right)=|W_{\psi }Z\left[t\left(s\left(\lambda \right)\right),s\left(\lambda \right)\right]|$ for an arbitrary trajectory t(s(λ)) as a volatility proxy. The quantity x(λ) is believed to be a generalization of empirical volatility, whereas the wavelet transform $\left|W_{\psi }Z[u,s]\right|$ is exactly the absolute value of the logarithmic return when we use ψ(t)=δ(t−1)−δ(t). The analyzing wavelet is selected to remove local trends from the original non-stationary time series. Since higher-order derivatives of the Gaussian function yield the same scaling function, the results are guaranteed to coincide at the zeroth-order approximation. For similar reasons, methods such as DFA (Detrended Fluctuation Analysis) [51] and DMA (Detrended Moving Average) [52] are also considered effective in removing local trends.

Figure 1 shows the probability density function (PDF) of the absolute values of wavelet coefficients on several time scales for data set 1. The longest time scale in Figure 1 is 1024 min, which corresponds to approximately two trading days on the LSE, while the shortest time scale is 8 min. The dashed line represents a Gaussian distribution, and it can be observed that the deviation in the tail region increases as the time scale decreases. This effect is called a fat tail, indicating that the impact of large intermittent price fluctuations cannot be neglected.

#### 2.4.4. Wavelet Transform Modulus Maxima Line

To verify the validity of the model (Equation 3) using actual data, the trajectory must be identified. A natural trajectory within the half-plane $S^{+}$ is the wavelet transform modulus maxima line (WTMML), which is a curve connecting the maxima of the wavelet transform modulus across different time scales [53,54,55]. These maxima can be regarded as the “ridges” of the wavelet transform modulus. For the mathematical definition, see Appendix B.

Figure 2 illustrates the original time series Z(t) and the WTMML extracted from this time series for data set 1. The cascade of volatility occurs from top to bottom. Therefore, sections of the WTMML moving to the right indicate a causal cascade, while sections moving to the left indicate an anti-causal cascade. This point will be discussed later.

## 3. Results

### 3.1. Parameter Estimation Based on Integral Fluctuation Theorem

The parameters $\gamma_{M}$, $\sigma_{M}$, *a*, and *b* are estimated using the approximate expression (Equation 7), relations (Equation 12) and (Equation 13) for data set 1, and the Integral Fluctuation Theorem (Equation 33). If we set ϵ=a as the control parameter and use relations (Equation 12) and (Equation 13), we obtain the relationships $\gamma_{M}=\sigma_{M}^{2}/2+0.5+\epsilon /A$ and $b=\sqrt{2A\epsilon }$. We then optimize the parameter ϵ so that the entropy production along the trajectory satisfies the Integral Fluctuation Theorem (Equation 33). We use relations (A3)∼(A6) instead of relations (Equation 12) and (Equation 13) for data sets 2 and 3.

The value of the parameter $\sigma_{M}^{2}\left(\epsilon \right)$ is also unknown. In [35], parameter estimation was carried out using the conditional moments $E[\left(x_{2}-x_{1}\right)|x_{1}]$ and $E[{\left(x_{2}-x_{1}\right)}^{2}|x_{1}]$. However, this approach results in large errors when sufficient data are not available and thus was not adopted here. Instead, we used the following simplified method. By comparing the expression for the scaling exponent $\zeta \left(q\right)=\gamma_{M}q-\frac{1}{2}\sigma^{2}q^{2}$ for the case a=b=0 with the empirical result $\tau \left(q\right)=\gamma_{M}q-\frac{1}{2}\sigma^{2}q^{2}-1$ reported in Figures 2 and 11 in [35], we fix $\sigma_{M}^{2}\left(\epsilon \right)≃\sigma_{M}^{2}\left(\epsilon =0\right)=0.026$. Thus, the parameters are not fully optimized.

Figure 3 illustrates the results of two types of parameter estimation. The maximum time scale (initial state) of the trajectory is set to 512 min (approximately one trading day), and the minimum time scale (final state) is set to 32 min.

One case uses the WTMML obtained from the time series Z(t) as the trajectory, represented by white circles in Figure 3, while the other is derived from simulations of the discretized Langevin Equation (Equation 20). The simulations were performed using a simple Euler method, generating 10 sets of simulations with 10,000 trajectories for each value of ϵ. Black circles indicate the mean, and error bars represent the standard deviation of the 10 sets of simulations.

The optimal value of ϵ is found to be ϵ=0.44 when using the WTMML as the trajectory and ϵ=0.32 when using the simulations for data set 1, as shown in Figure 3. Although there is a slight discrepancy since only 113 WTMMLs pass the maximum time scale of 512 min within the data interval (*L* =131,072 min), this difference should be considered to be within an acceptable range. The optimal value of ϵ is found to be ϵ=0.067(0.00015) when using WTMML as the trajectory and ϵ=0.066(0.00015) when using the simulations for data set 2 (Mitsubishi UFJ Financial Group in data set 3). The differences are small enough. These findings support the validity of the concept of WTMMLs for representing the trajectories of the volatility cascade.

Figure 4 shows the PDF of the entropy generated in the entire system, including the heat bath, during the transition from the initial state to the final state for data set 1.

Figure 5 shows the PDF of |x|, on several time scales, obtained from simulations using each optimal value of the parameter ϵ for each data set. The initial state was generated randomly according to the PDF of $\left|W_{\psi }Z[,s]\right|$ at s=512 min. The simulation results are represented by black circles and solid lines, while the actual PDF of $\left|W_{\psi }Z[,s]\right|$ is represented by red circles. Overall, the simulation results successfully reproduce the actual PDF.

### 3.2. Details of Trajectories

Let us examine the average behavior of the thermodynamic quantities for each trajectory. Figure 6 shows the results of data set 1 for the entropy consumption trajectories in which the total entropy productions are negative, while Figure 7 shows the results for the entropy production trajectories in which the total entropy productions are positive.

We interpret the variation in volatility along the trajectory as analogous to the motion of a Brownian particle immersed in water (a heat bath). In financial markets, we consider the role of the heat bath to be played by latent market participants.

Financial markets consist of various participants who observe the market with different temporal resolutions and react to price changes over different time horizons. Competition among participants with different characteristic time horizons gives rise to a heterogeneous structure of volatilities described by multifractality.

The panels in Figure 6 and Figure 7 illustrate the average variations in the following thermodynamic quantities along the trajectory from a longer to a shorter time scale.

**(A)** **Volatility:** This represents the position of the Brownian particle, which corresponds to the system’s state variable.**(B)** **Force applied to the system:** This indicates the external force acting on the Brownian particle.**(C)** **Inverse temperature of the heat bath:** According to Equations (16) and (26), the temperature represents the trajectory of the variance of volatility.**(D)** **Heat transfer:** This describes the energy transferred from the heat bath to the system, that is, the energy received by the Brownian particle from the heat bath. A negative value indicates a positive amount of heat flowing from the particle back to the heat bath.**(E)** **Change in the content of the information of the system:** As expressed in Equation (Equation 32), this quantity can be interpreted as the change in the entropy of the system, a central concept in stochastic thermodynamics.**(F)** **Change in total entropy:** This indicates the change in the total entropy of the system and the heat bath.

The left panel of Figure 6 for each thermodynamic quantity (A)∼(F) represents the average of six entropy consumption trajectories of a total of one-hundred-thirteen WTMMLs, while the right panel represents the average of the top one-hundred cases with the largest negative entropy production values of the simulated trajectories.

In the case of the entropy consumption shown in Figure 6, a common feature between the left and right panels is that most thermodynamic quantities start to exhibit large fluctuations around the time scale of 100 min. An exception is the inverse temperature β, which begins to increase around the time scale of 200 min.

A key difference between the left and right panels is that, in the WTMML, all the thermodynamic quantities except the inverse temperature β exhibit maxima or minima around the time scale of 200 min, whereas this behavior is not observed in the simulated trajectory.

In the case of the entropy production shown in Figure 7, the left and right panels exhibit almost the same fluctuation trend.

Next, we examine the profile of WTMMLs within the half-space $S^{+}$. In Figure 8, all the WTMMLs have been shifted in the time direction so that their positions on the 512-minute time scale align with the origin of the time axis. Panels (A), (B), and (C) represent the average positions of WTMMLs on each time scale for the entropy consumption trajectories, the entropy production trajectories, and all the trajectories, respectively. The three columns, from left to right, represent the results of data sets 1 and 2 and the aggregated results of the trajectories of the seven stocks in data set 3, respectively.

Since the trajectories in all the panels move from top to bottom, either rightward or nearly vertical, it can be seen that the volatility cascade occurs at a nonuniform speed along the time axis for data set 1. This observation is consistent with the stylized fact in financial time series that volatility at coarse time scales predicts volatility at finer time scales. In contrast, for the two data sets from the TSE, while similar behavior is observed at smaller time scales, anti-causal behavior emerges at longer time scales. From the results of data set 2, it can be seen that an anti-causal section exists between the time scales of the range from 100 to 200 min, except for the entropy consumption trajectory, which involves only a small number of data points (n=5). In these anti-causal sections, it is considered that the reversed process of the Langevin equation is being realized along the causal time direction. In addition, the aggregated trajectories of the seven stocks in data set 3 indicate that, for the entropy production trajectory, the region corresponding to time scales longer than 200 min exhibits anti-causal behavior. Although neither the LSE nor the TSE data sets were examined for time scales longer than 512 min, the differences observed below 512 min may be attributed to the interruption in trading hours caused by the lunchtime intermission on the TSE. We intend to investigate these points further using larger data sets, including the region beyond the 512-minute time scale.

## 4. Discussion

We study a continuous multiplicative random cascade model of volatility cascade processes formulated as a stochastic differential equation in the framework of stochastic thermodynamics by considering it as a Langevin system. We introduce thermodynamic concepts such as temperature, heat, work, and entropy into the analysis of financial markets. This framework allows for a detailed investigation of individual trajectories of volatility cascades across different time scales, from longer to shorter. The model includes two independent modes of Brownian motion: one has multiplicative coupling with volatility; the other has additive coupling.

The model parameters can be estimated based on the system’s multifractality and the expression of an Integral Fluctuation Theorem derived by applying the formalism of stochastic thermodynamics to the Langevin system. The Brownian process additively coupled to volatility is found to be not only consistent with the empirical observations discussed in Refs. [29,35] but also a fundamentally essential component for applying the Langevin equation to financial systems.

An empirical analysis was conducted using volatility extracted via wavelet transform from the time series of the logarithmic stock prices of the three data sets from different stock markets, the LSE and TSE; it showed that, by appropriately setting the control parameter, the Langevin equation successfully reproduced the PDF of volatility at each time scale.

Additionally, the working hypothesis that WTMMLs can be used to identify the trajectories represented by the Langevin system within the half-plane of time and time scale was found to be reasonably accurate given the limited number of observed trajectories. These trajectories were confirmed to be causally structured on average for data set 1, reaffirming the stylized fact in financial time series that past coarse-grained measures of volatility correlate better to future fine-scale volatility than the reverse process does. In contrast, for the two data sets from the TSE, while similar behavior is observed at smaller time scales, anti-causal behavior emerges at longer time scales.

In future research, we plan to conduct an empirical analysis with a significant increase in the number of samples used to further validate these findings. As an application to investment strategies and risk management, we believe that estimating model parameters and predicting volatility cascade trajectories over a given target period can enable detailed risk forecasting. To achieve this, it is necessary to extend the present methodology to longer time scales, such as weekly or monthly, and to establish methods beyond WTMML to identify and forecast volatility cascade trajectories. The implications of the evolution of thermodynamic quantities along these trajectories in financial markets are not yet fully understood. We expect that comparing them with the temporal evolution of order books may offer new insights [56]. The existing models, such as the GARCH class and stochastic volatility models, are temporal evolution models defined at fixed time horizons. The construction of a temporal evolution model based on the Langevin equation, as well as the investigation of its relationship with these existing models, remains a topic for future research. In this context, the volatility cascade from long- to short-term scales represents a process that breaks time-reversal symmetry. Similarly, asymmetric phenomena such as price–volatility correlation, often referred to as the leverage effect, represent important issues. The relationship of the proposed model with information thermodynamics is also expected to be a highly interesting direction for further exploration [57].

## Figures and Tables

**Figure 1 entropy-27-00435-f001:**
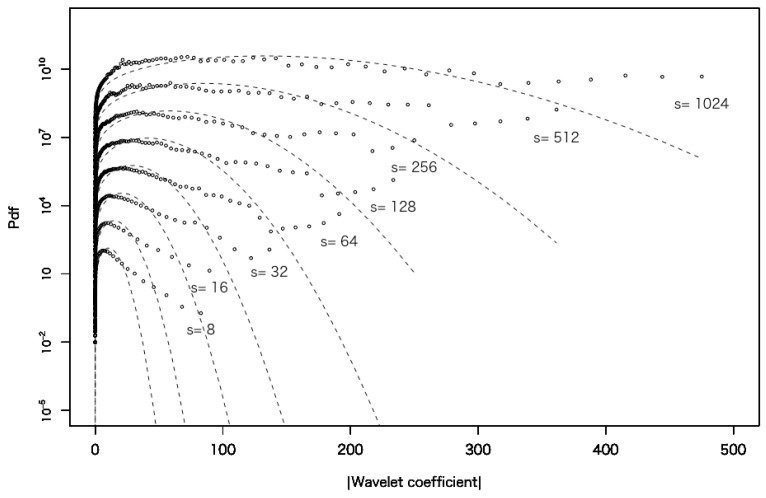
The PDFs of the absolute values of the wavelet coefficients on several time scales for data set 1. The time scale is indicated next to each curve. The Gaussian distribution with the same mean and variance is shown as a dashed line. Each curve is vertically shifted for better readability.

**Figure 2 entropy-27-00435-f002:**
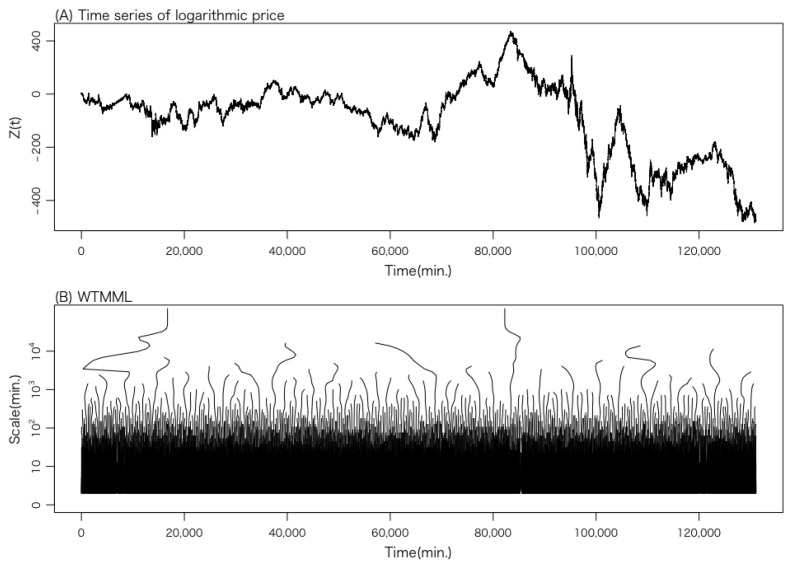
The original time series Z(t) and the WTMML extracted from this time series for data set 1. (**A**) The time series Z(t) of logarithmic price. (**B**) The corresponding WTMML.

**Figure 3 entropy-27-00435-f003:**
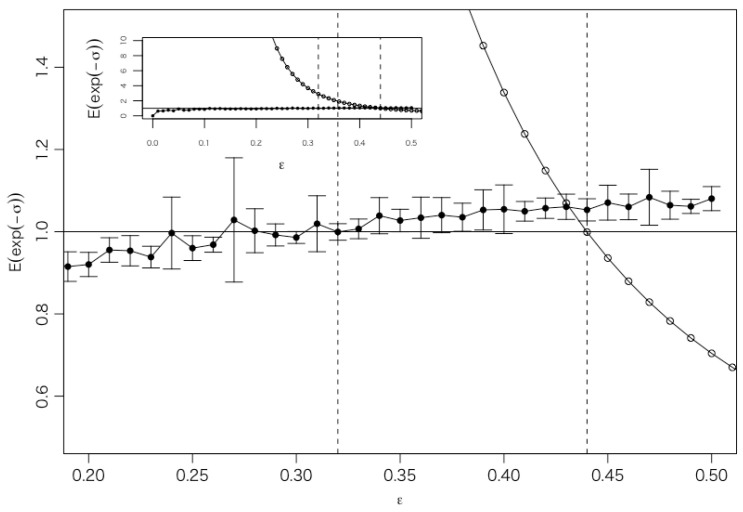
The value of E(exp(−σ)) against control parameter ϵ for data set 1. One case uses the WTMML as the trajectory, represented by white circles. The other case is derived from simulations of the discretized Langevin Equation (Equation 20). The simulations were performed using a simple Euler method, generating 10 sets of simulations with 10,000 trajectories for each value of ϵ. Black circles indicate the mean, and error bars represent the standard deviation of the 10 sets of simulations. The plot in the inset shows the value of E(exp(−ϵ)) over a broader range of ϵ. The optimal value of ϵ is found to be ϵ=0.44 when using the WTMML as the trajectory and ϵ=0.32 when using the simulations.

**Figure 4 entropy-27-00435-f004:**
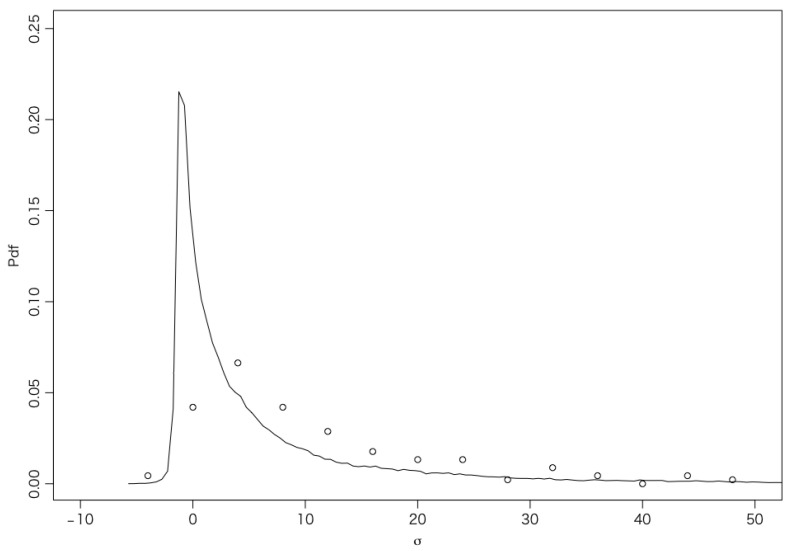
The PDF of the entropy generated in the entire system for data set 1. The white circles represent the production of entropy in the WTMML using the control parameter ϵ=0.44. The mean entropy production is E[σ]=10.4. The solid line represents the results of 100,000 simulations with the control parameter ϵ=0.32, where the mean entropy production is E[σ]=5.8.

**Figure 5 entropy-27-00435-f005:**
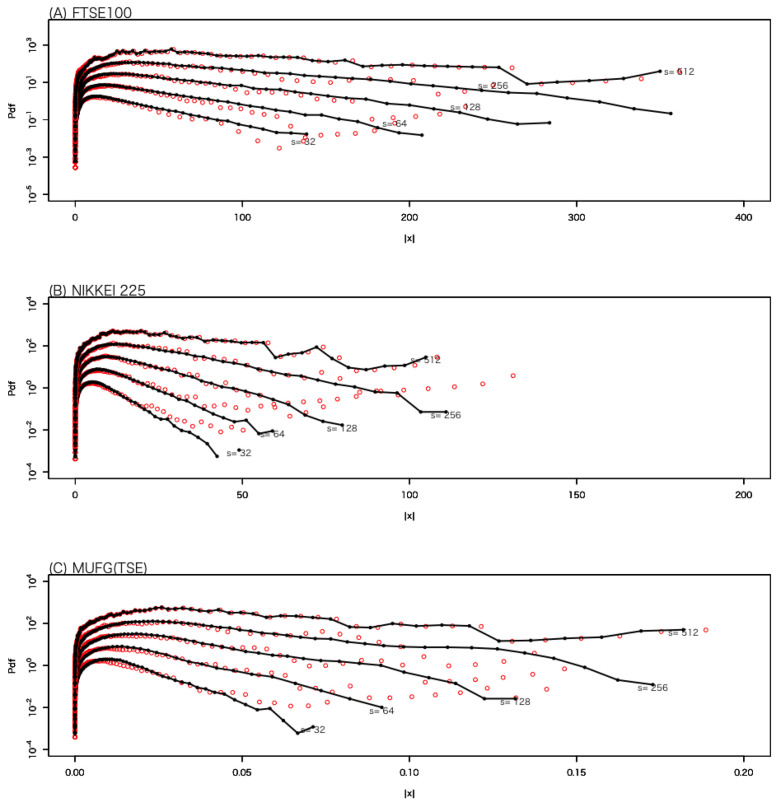
The PDF of |x|, on several time scales, obtained from simulations using each optimal value of the parameter ϵ for each data set. The initial state was generated randomly according to the PDF of $\left|W_{\psi }Z[,s]\right|$ at s=512 min. The simulation results are represented by black circles and solid lines, while the actual PDF of $\left|W_{\psi }Z[,s]\right|$ is represented by red circles.

**Figure 6 entropy-27-00435-f006:**
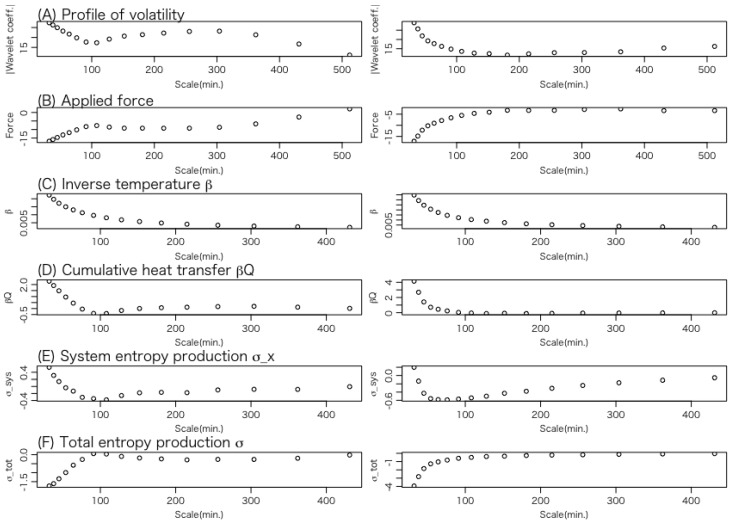
The average behavior of thermodynamic quantities (**A**–**F**) of data set 1 for each trajectory in the case of entropy consumption. Thermodynamic quantities (**D**–**F**) are cumulative quantities. The left panel of the figure for each of the thermodynamic quantities (**A**–**F**) represents the average of 6 negative entropy production trajectories of a total of 113 WTMMLs, while the right panel represents the average of the top 100 cases with the largest negative entropy production values of the simulated trajectories.

**Figure 7 entropy-27-00435-f007:**
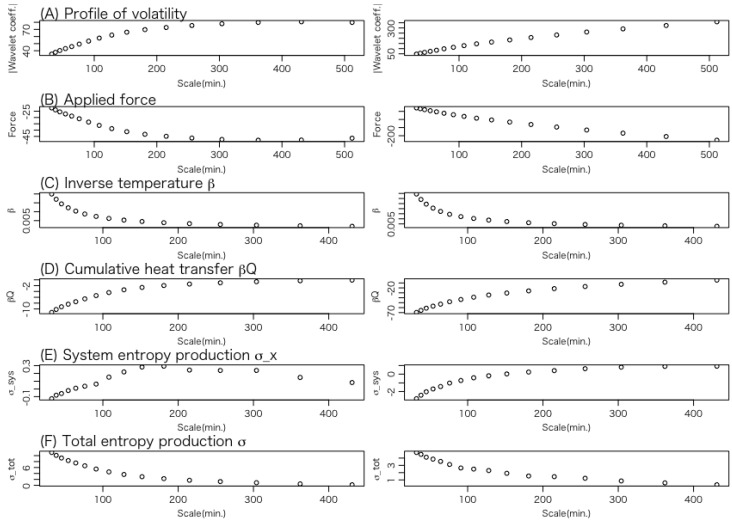
The average behavior of thermodynamic quantities (**A**–**F**) of data set 1 for each trajectory in the case of entropy production. Thermodynamic quantities (**D**–**F**) are cumulative quantities. The left panel of the figure for each of the thermodynamic quantities (**A**–**F**) represents the average of 107 positive entropy production trajectories of a total of 113 WTMMLs, while the right panel represents the average of the top 100 cases with the largest positive entropy production values of the simulated trajectories.

**Figure 8 entropy-27-00435-f008:**
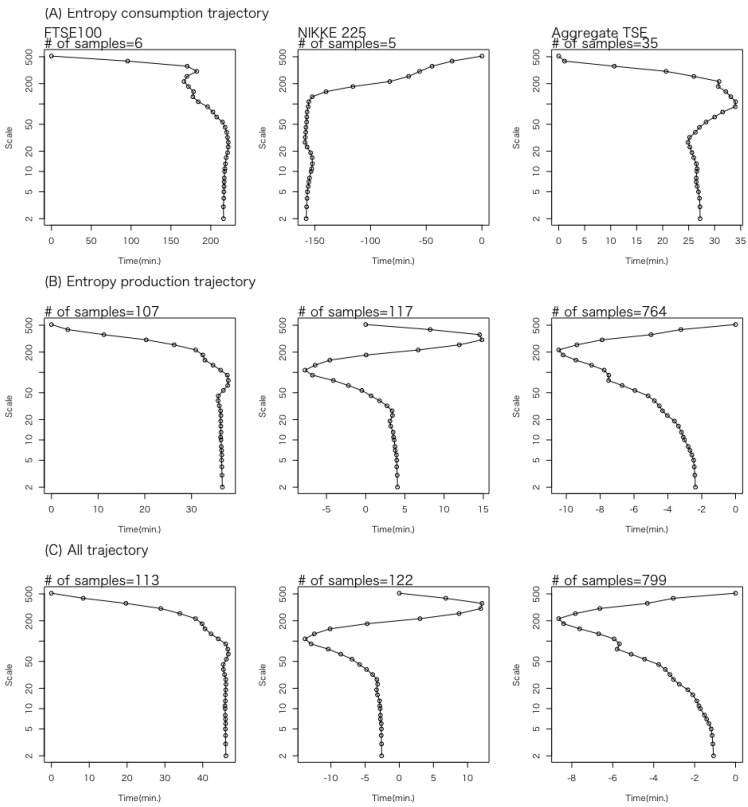
The profile of WTMMLs within the half-space $S^{+}$. All WTMMLs have been shifted in the time direction so that their positions on the 512-minute time scale align with the origin of the time axis. Panels (**A**–**C**) represent the average positions of WTMMLs on each time scale for the entropy consumption trajectories, the entropy production trajectories, and all the trajectories, respectively. The three columns, from left to right, represent the results of data sets 1 and 2 and the aggregated results of the trajectories of the seven stocks in data set 3, respectively.

## Data Availability

Publicly available data sets were analyzed in this study. The LSE data can be found at the LSE Historic Price Service (HPS) (https://www.londonstockexchange.com/products-and-services/reference-data/hps/hps.htm, accessed on 28 February 2025). The TSE data can also be obtained by contacting Nikkei Needs (needs@nikkei.co.jp).

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
