# Peer review of "Empirical Study on Fluctuation Theorem for Volatility Cascade Processes in Stock Markets"

_entropy, 2025, doi:10.3390/e27040435_

Round 1

Reviewer 1 Report

Comments and Suggestions for Authors

Referee Report on: "Empirical Study on a Fluctuation Theorem for Volatility Cascade Processes in Stock Markets"

Manuscript ID: entropy- 3529629

The study explores volatility cascade processes through the framework of stochastic thermodynamics. While this is an interesting angle, I find several fundamental issues with the manuscript that make it difficult to recommend for publication in its current form.

The primary concern is the lack of a clear contribution to financial economics. The paper is heavily theoretical and focused on stochastic thermodynamics, but it does not convincingly explain how this framework improves our understanding of financial markets. The motivation appears to be more aligned with statistical physics than with the concerns of financial economists, policymakers, or practitioners. Without a stronger connection to empirical finance, it is unclear why this approach is relevant to the journal’s readership.

The literature review is also (really) outdated. The paper does not engage with recent developments in volatility modeling, nor does it compare its findings to standard methods used in financial market research. I suggest incorporating more recent studies in the field, such as [ https://doi.org/10.1111/mafi.12354]. These references could help position the paper within the broader financial economics literature.

Beyond these issues, the manuscript lacks clarity in its research question and overall structure. The empirical results are difficult to interpret, and it is unclear what practical or theoretical insights emerge from the study. The discussion of policy or real-world implications is absent, further limiting the paper’s relevance to a wider audience.

I encourage the authors to reconsider the scope and positioning of their research to align more closely with the interests of the finance and economics community.

Author Response

I thank Reviewer #1 for reviewing my manuscript and providing me insightful comments, I have addressed them accordingly in the manuscript, and also respond to the comments below:

1) The primary concern is the lack of a clear contribution to financial economics. The paper is heavily theoretical and focused on stochastic thermodynamics, but it does not convincingly explain how this framework improves our understanding of financial markets. The motivation appears to be more aligned with statistical physics than with the concerns of financial economists, policymakers, or practitioners. Without a stronger connection to empirical finance, it is unclear why this approach is relevant to the journal’s readership.

As suggested by Reviewer #1, I have added some paragraphs in Abstract and Introduction to explain the reason why we apply the stochastic thermodynamics to financial market and the results that can only be obtained in that way.

2) The literature review is also (really) outdated. The paper does not engage with recent developments in volatility modeling, nor does it compare its findings to standard methods used in financial market research. I suggest incorporating more recent studies in the field, such as [ https://doi.org/10.1111/mafi.12354]. These references could help position the paper within the broader financial economics literature.

Following the proposal of Reviewer #1, I include the suggested reference in Introduction to position this study within the context of existing financial time series models.

3) Beyond these issues, the manuscript lacks clarity in its research question and overall structure. The empirical results are difficult to interpret, and it is unclear what practical or theoretical insights emerge from the study. The discussion of policy or real-world implications is absent, further limiting the paper’s relevance to a wider audience.

For this Reviewer #1’s concern, I added the paragraph to the Introduction to make the research question clearer, and the explanation to the Result to make it easier to understand the meaning of the results. Furthermore I has expanded the discussion to interpret the implications of the findings in the context of financial time series research.

Reviewer 2 Report

Comments and Suggestions for Authors

Dear Authors

It was a pleasure reading your work. It is an interesting study but requires minor improvements before it can be accepted for publication to maximize the article's impact.

Please find them in the attached file.

All the best.

Comments on the Quality of English Language

The English language is appropriate and understandable.

Author Response

I thank Reviewer #2 for reviewing my manuscript and providing me insightful comments, I have addressed them accordingly in the manuscript, and also respond to the comments below:

1) The keywords chosen for the study focus but could include “financial

thermodynamics” or “econophysics” for better indexing.

Following the proposal of Reviewer #2, I include “econophysics” as the Keyword of the article.

2) Please include more recent research on stochastic thermodynamics applied to financial markets. Add references from the last five years on stochastic volatility models and applications of the Langevin equation in finance.

According to the comment of Reviewer #2, I include some recent references relevant to my work.

3) Furthermore, the introduction strongly focuses on the chosen approach but does

not adequately discuss competing models or why the chosen framework is

superior. To justify the selection, try to compare the Langevin-based approach

with other models used to describe financial time series, such as GARCH or

multifractal models.

As suggested by Reviewer #2, I have added some paragraphs in Abstract and Introduction to explain the reason why we apply the stochastic thermodynamics to financial market and the results that can only be obtained in that way.

4) Finally, it is stated in this section that volatility cascades in

financial markets can be studied using stochastic thermodynamics. However, it

is not explicitly justified why this is a suitable framework. Please explain

explicitly how entropy production and the fluctuation theorem contribute to

understanding volatility cascades.

According to the remark of Reviewer #2, I add the paragraph In the Introduction Page 2 beginning with the line 60 to explain the contribution of the fluctuation theorem.

5) Address whether similar results hold in more stable market conditions.

Why to select the FTSE100 Index?

According to the remark of Reviewer #2, I executed some additional analyses for two data sets.

6) Justify this choice and consider whether different wavelets yield similar results.

According to the remark of Reviewer #2, I add the paragraph In the Materials and Methods Page 8 beginning with the line 243 to address the concern.

7) However, this section also requires improvements. First, it lacks a critical

evaluation of the model's limitations. Please acknowledge possible weaknesses,

such as Langevin equation assumptions or parameter tuning sensitivity.

Second, explain how these findings could inform trading strategies, volatility

forecasting, or risk management. Third, the discussion does not compare the

effectiveness of this approach against traditional financial models like GARCH

or stochastic volatility models. Thus, briefly discuss how this method differs from

and potentially improves upon existing models. Finally, the discussion ends

without a strong concluding statement summarizing the key contributions.

Clearly state the main findings, their significance, and the next steps for future

research.

According to the remark of Reviewer #2, I add some paragraphs in Discussin Page 16 beginning with the line 398 to address the concerns.

Reviewer 3 Report

Comments and Suggestions for Authors

In this paper, Prof Maskawa incorporated the integral fluctuation theorem and entropy production into the stochastic differential equation (SDE) framework developed in [Front. Phys. (Lausanne), 2020, 8, 565372]. Using this new SDE framework, as well as FTSE 100 constituents on London Stock Exchange as time series data from Nov 2007 to Jan 2009, he was able to reproduce the empirical volatility distribution profile, as well as to reaffirm the stylized fact of a causal structure when volatility cascades from large scales to small scales. In addition, the scaling characteristics of some thermodynamic quantities, e.g., inverse temperature, heat transfer, and entropy production with time scale are also presented for two different simulated trajectories, one consuming entropy, and the other producing entropy. In this way, he elucidated how thermodynamic properties of a complex system vary with time scale.

Volatility clustering is a stylized fact well known in economics, finance, and econophysics. The author used a modified SDE framework to simulate how volatility cascades over different time scales, facilitating a deeper understanding of its mechanisms. The new framework also opens new research directions, such as spatial-temporal phenomena and dynamics.  Additionally, entropy production is a robust technique for investigating the dynamical properties of nonlinear, non-stationary, and open-systems. The inclusion of this part to the SDE framework is original and inspire researcher on how to engage in entropy-related topics and thermal properties of complex systems, such as financial markets and biological systems. Overall, the new SDE framework with integral fluctuation theorem and entropy production and the findings in this work is novel and impactful, and we recommend that the manuscript be accepted for publication in Entropy. However, we have a few major concerns (along with some minor comments) that we would like the author to address before the manuscript is published.

 Our major concerns are:

  1. In [ Front. Phys. (Lausanne),2020, 8, 565372], the authors mentioned the casual structure from long-term to short-term volatility, and the price-volatility correlation is both related to symmetry breaking in the time axis. Nevertheless, the author did not mention if using this new SDE framework, they can also reproduce the latter stylized fact in real market or not. We suggest author to give brief remarks on this in the Discussion section.
  2. The total length of financial time series is 140,000 minutes. However, the period which the author used to test the stylized fact ‘past coarse-grained measures of volatility correlate better to future fine-scale volatility than the reverse process does’ is only limited to t = 520 min to  t = 600 min, which is quite a short period compared to the total period. We worry that their results and conclusion might therefore be biased. To confirm this stylized fact more robustly, it is better to include more periods, for example, the author can do a similar analysis for the 2009 Global Financial Crisis crash period, so that we can know the difference between volatility cascades in normal times, and during market crashes. In addition, there is a range of time scale for the same period  t = 520 min to t = 600 min, which is from T_scale from 1000 to 8000 min, where we find an anti-casual cascade. In fully-developed turbulence, the dominant process is large eddies breaking up into smaller eddies breaking further up onto even smaller eddies. This is why scientists hunt for causal structures consisting of cascades going from long time scales to short time scales. In a stock market, the reverse process is also possible, with smaller structures coalescing into larger scales. We would like the author to discuss whether this reverse process constitutes the same stylised fact as the cascade from large to small, or could it represent a new stylised fact not previously reported.
  3. We suggest the author to give a concise introduction to the old SDE framework they established in [ Front. Phys. (Lausanne),2020, 8, 565372], and contrast the new changes he made in this new SDE framework. He can either put it in the Introduction section, or in the Materials and Methods section. Also, the author should describe what the hypothesis and research questions are in the last paragraph of the introduction, and connect what the author has found and achieved in this work back to them, so readers can follow closely on what the new SDE framework resolved and achieved.
  4. In Figure 5, the author computed volatility distribution at different time scales. We think it is necessary to provide two additional such results for ε slightly less than 0.32 and ε slightly greater than 0.32, and confirm if the fitting errors of (1) the volatility distribution profile with respect to empirical data and, (2) integral fluctuation theorem into account is still visibly optimal at ε = 0.32. If the optimal value is different from ε = 0.32, the author should give some remarks in the Discussion section.

We also have the following minor comments:

  1. In the Acknowledgments section, “He also would like to express my gratitude for the fruitful discussions with Professor Emeritus Kuroda of Nihon University.” shall be changed to “He also would like to express my his gratitude for the fruitful discussions with Professor Emeritus Kuroda of Nihon University.”
  2. In Figure 8(c), the x-axis label shall be Time(min) instead of u.
  3. In Figure 7 caption, “The average behavior of the thermodynamic quantities (A) ∼ (F) for each trajectory in the case of the entropy production. The explanation is the same as that of Figure” shall be changed to “The average behavior of the thermodynamic quantities (A) ∼ (F) for each trajectory in the case of the entropy production. The explanation is the same as that of Figure 6”.

Author Response

I thank Reviewer #3 for reviewing my manuscript and providing me insightful comments, I have addressed them accordingly in the manuscript, and also respond to the comments below:

1) In [ Front. Phys. (Lausanne),2020, 8, 565372], the authors mentioned the casual structure from long-term to short-term volatility, and the price-volatility correlation is both related to symmetry breaking in the time axis. Nevertheless, the author did not mention if using this new SDE framework, they can also reproduce the latter stylized fact in real market or not. We suggest author to give brief remarks on this in the Discussion section.

According to the suggestion of Reviewer #3, I add the paragraph in Discussin Page 16 beginning with the line 411 to state as the future research issue.

2) In fully-developed turbulence, the dominant process is large eddies breaking up into smaller eddies breaking further up onto even smaller eddies. This is why scientists hunt for causal structures consisting of cascades going from long time scales to short time scales. In a stock market, the reverse process is also possible, with smaller structures coalescing into larger scales. We would like the author to discuss whether this reverse process constitutes the same stylised fact as the cascade from large to small, or could it represent a new stylised fact not previously reported.

This is the essentially important comment for me. I executed some additional analyses according to feedback from reviewers, and find anti-causal section of trajectories. It is mentioned in Results and Discussion.

3) We suggest the author to give a concise introduction to the old SDE framework they established in [ Front. Phys. (Lausanne),2020, 8, 565372], and contrast the new changes he made in this new SDE framework. He can either put it in the Introduction section, or in the Materials and Methods section. Also, the author should describe what the hypothesis and research questions are in the last paragraph of the introduction, and connect what the author has found and achieved in this work back to them, so readers can follow closely on what the new SDE framework resolved and achieved.

According to the remark of Reviewer #3, I add the paragraph in Introduction Page 2 beginning with the line 68 and 86.

4) In Figure 5, the author computed volatility distribution at different time scales. We think it is necessary to provide two additional such results for ε slightly less than 0.32 and ε slightly greater than 0.32, and confirm if the fitting errors of (1) the volatility distribution profile with respect to empirical data and, (2) integral fluctuation theorem into account is still visibly optimal at ε = 0.32. If the optimal value is different from ε = 0.32, the author should give some remarks in the Discussion section.

According to the remark of Reviewer #3, I put a comment in Result Page 9 beginning with the line 273.

Round 2

Reviewer 2 Report

Comments and Suggestions for Authors

Dear Authors,

Thank you for your efforts in revising the manuscript. Your careful revisions have significantly improved the manuscript, making it a strong contribution to the field. I appreciate your dedication to refining the work and ensuring its clarity and scientific rigor.

Best regards,

Author Response

Thank you for carefully reading my paper. Your various suggestions have greatly improved the quality of the paper.

Reviewer 3 Report

Comments and Suggestions for Authors

The author has replied to all my major concerns satisfactorily. In particular, the author has collected two additional data sets, i.e. Nikkei 225, and MUTG(TSE), and computed volatility distribution profile, wavelet transform modulus maxima line (WTMML) trajectories, and thermodynamical quantities for the new data set. Also, the data set covered different periods, where the volatility cascades are computed, making the confirmation of volatility casualty less biased. Finally, the anti-causal cascades are also investigated and discussed.

After fixing the following minor issues, the revised manuscript can be published in Entropy. 

  1. Line 131 FIGURE 1 => Figure 1
  2. Acknowledgments:.. He also expresses my gratitude for the fruitful discussions with Professor Emeritus Kuroda of Nihon University,…” should be “He also expresses my his gratitude for the fruitful discussions with Professor Emeritus Kuroda of Nihon University,…”
  3. Figure 8(A) NIKKE$225 => NIKKEI 225
  4. Figure 7(e) system entropy production _x =>
  5. Line 220 => NIKKEI225 => Nikkei 225
  6. Figure 5(b) NIKKEI225 => NIKKEI 225

Author Response

(The authors gave the same response as above.)
